# WHAT SHOULD AN AI ASSESSOR OPTIMISE FOR?

## ABSTRACT

An AI assessor is an external, ideally independent system that predicts an indicator, e.g., a loss value, of another AI system. Assessors can leverage information from the test results of many other AI systems and have the flexibility of being trained on any loss function: from squared error to toxicity metrics. Here we address the question: is it always optimal to train the assessor for the target loss? Or could it be better to train for a different loss and then map predictions back to the target loss? Using ten regression problems with tabular data, we experimentally explore this question for regression losses with monotonic and nonmonotonic mappings and find that, contrary to intuition, optimising for more informative losses is not generally better. Surprisingly though, some monotonic transformations, such as the logistic loss used to minimise the absolute or squared error, are promising.

## 1 INTRODUCTION

AI models and systems are evaluated with very different metrics, depending on the purpose of application. For instance, metrics as diverse as the BLEU score (Papineni et al., 2002) for translation, 'Bold' toxicity score (Dhamala et al., 2021) for text generation, the area under the ROC curve (Fawcett, 2006) for classification, asymmetric loss (Elliott et al., 2005) for sales prediction (Gogolev & Ozhegov, 2023) or any reward function (Eschmann, 2021) for reinforcement learning, are commonly used. Models can be built or trained to minimise some loss, and then repurposed for a situation where another metric matters more. The most characteristic example today of this process is represented by 'foundation models' (Bommasani et al., 2021), such as language models. Even if the model can produce uncertainty estimates about the next token, and these are well calibrated, the metric of interest may be toxicity. Since the model does not estimate toxicity, we need some external way to do this.

One solution to this challenge is the development of *assessor models* (Hernández-Orallo et al., 2022). An assessor is a predictive model designed to estimate how well another system, called the base or subject system $s$, will perform on a given example or problem instance $i$ for a specific validity metric before it is actually deployed. An assessor can estimate the conditional distribution $\hat{p}(v|s,i)$ or simply (pointwise) map $\langle s,i \rangle \mapsto v$. Assessors are related to verifiers (Li et al., 2023) but are *anticipatory*: rather than simply checking outcomes post-execution, they predict the outcomes in advance (i.e., given a new example $i$, they can predict the value $v$ of the metric that $s$ is expected to achieve). For instance, consider $s$ a self-driving car and $i$ a specific journey. An assessor could predict the safety outcome $v$ of $s$ for $i$.

Assessors are used to anticipate any metric of quality, safety, bias or, in general, validity for any kind of subject system, from RL agents to language models. Assessors can be used to monitor or forecast system performance (Schellaert et al., 2024), to optimise configurations (Zhao et al., 2024), to do anticipatory reject (Zhou et al., 2022; da Costa et al., 2023), or to delegate by routing(Hu et al., 2024; Lu et al., 2023; Ding et al., 2024). Assessors are usually trained on test data, capitalising on vast information from results of many systems and examples (Burnell et al., 2023).

It may seem natural that the assessor is trained to optimise for the metric we are interested in. For instance, if the subject system $s$ estimates daily energy consumption of households and the metric value $v$ is given by the squared error ($L_2^+$) between actual and estimated consumption values, then one would expect that the assessor should be trained to predict the squared error that the system will incur for each household. However, in this paper *we challenge the general assumption that training*

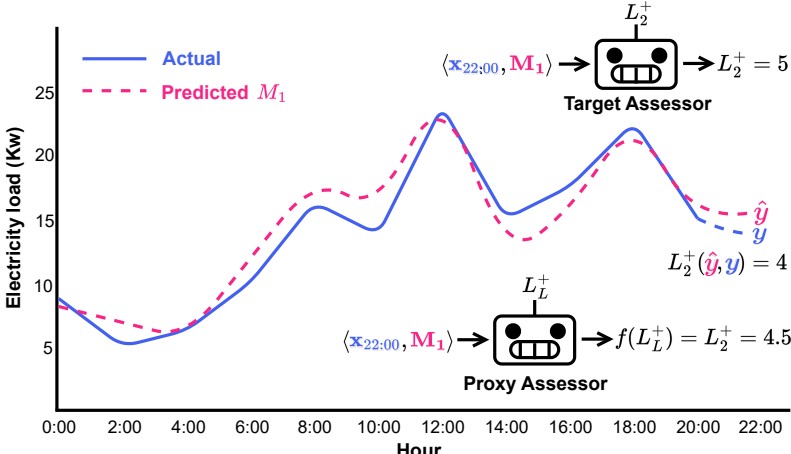

Figure 1: For an energy consumption model $M_1$, we want to anticipate the squared error ($L_2^+$) for each new example using an external predictor, called assessor. Recommendations to customers are only made when the assessor predicts low squared error in the energy consumption estimate. In this paper we explore assessors that optimise for the target loss function (squared loss $L_2^+$, top) but also assessors that use a proxy loss function (logistic loss $L_L^+$, bottom) followed by a transformation ($f$). Can the proxy assessor be better?

*an assessor to optimise directly for a specific metric $L$ necessarily results in the best optimisation outcome for $L$*. In this example, what if optimising for logistic loss ($L_L^+$) were better? This situation is illustrated in Figure 1.

To start exploring this question, in this paper we will consider the base model is solving a regression problem and we will use generic regression metrics, such as absolute error, squared error and logistic error. We will consider signed and unsigned (absolute) versions of these three metrics, and explore whether optimising for a proxy metric is better than optimising for the target metric. From our experimental analysis we observe some results that may be explained by the distribution of errors (residuals) in the test data of the base subject systems. However, some other results are more surprising, such as the logistic error being the best in all situations. This finding suggests that learning an assessor for one central metric might suffice to optimise a family of monotonically-related metrics.

## 2 BACKGROUND

This work situates itself within a broad spectrum of research on error analysis and the exploration of alternative loss functions for training predictive models. However, the use of assessors resituates this question at the meta-level, as a second-order regression problem, an area that, to our knowledge, has not been explored yet.

### 2.1 ERROR ANALYSIS IN REGRESSION

In regression problems, the choice and optimisation of loss functions is critical to model performance. There is an extensive literature on traditional error measures (Hyndman & Koehler, 2006; Botchkarev, 2018; 2019; Chicco et al., 2021) such as Mean Squared Error (MSE), Absolute Error, and more robust variants such as Huber Loss (Owen, 2007), which falls somewhat in between squared and absolute error, or Tukey's biweight loss (Beaton & Tukey, 1974; Belagiannis et al., 2015), which caps quadratic loss beyond a given point. Optimisation of these loss functions leads to different kinds of bias. For instance, quadratic error leads to estimators that are unbiased for the mean while absolute error leads to estimators that are unbiased for the median.

Beyond their use in performance evaluation, the analysis of errors and residuals also serves a diagnostic purpose, helping to identify model inadequacies or violations of assumptions, providing a

comprehensive understanding of the linear and non-linear relationships captured by regression models. For instance, Rousseeuw & Leroy (2005) use regression diagnostics, e.g., outlier diagnosis, to identify problems in both the explanatory and response variable, further refining the understanding of errors in predictive models.

Some studies have also explored more complex loss functions and their impact on regression model performance. According to Gneiting & Raftery (2007), appropriate scoring rules incentivise truthful prediction by optimising prediction distributions. However, as models and tasks become more complex, optimising a single loss function may not always align with the broader objectives of the system. In this regard, research such as (Huber, 1992) experiment with alternative, often non-convex, loss functions designed to improve model training under specific constraints or performance benchmarks.

## 2.2 ASSESSORS

The concept of *assessors* was first introduced in (Hernández-Orallo et al., 2022), and further explored specifically for large language models (LLM) by Zhou et al. (2022), who presented encouraging results in a limited setting involving a small domain focused on data wrangling. Kadavath et al. (2022) extended this by examining LLM and their role as assessors, finding that larger models tended to be more accurate and consistent in predicting outcomes across multiple tasks, although they acknowledged a lack of generalisation in out-of-distribution scenarios. Other applications of assessors focus on forecasting system performance (scaling laws) (Schellaert et al., 2024), team configurations (Zhao et al., 2024), anticipatory reject (Zhou et al., 2022; da Costa et al., 2023) or delegation (routing) to the best language model depending on the prompt (Lu et al., 2023; Hu et al., 2024; Ding et al., 2024). However, an analysis of the chosen validity metric and its distribution has not been done to date.

An assessor is an external, second-order system that predicts the scores of another, first-order system, the subject. It is populational, trained on test data spanning numerous instances and potentially multiple subjects. It operates as a standalone entity, independent of the subject. This attribute allows it to be anticipatory; it can predict the subject's performance solely on the basis of the input and the subject's characteristics, without needing access to the subject's output or the ability to execute it. Furthermore, the standalone nature of assessors offers advantages in terms of accountability and verification, as they can be developed by external auditors or for datasets different from those used to train the original subject. In addition, their use extends to increasing curriculum complexity, as in Bronstein et al. (2022), or facilitating instance-level model selection, a concept derived from algorithm selection (Kerschke et al., 2019). Finally, a perfect assessor (in an ideal scenario) would completely capture the epistemic uncertainty (error) associated with the subject's performance (Hüllermeier & Waegeman, 2021), with the error of the assessor depending only on the aleatoric error of the subject.

Assessors must learn from a very specific kind of distribution, given by the results of a loss function applied to the predictions of the base model. For instance, if this loss function is based on residuals, the dependent variable in the regression problem the assessors have to deal with will be affected by the distribution of residuals. Depending on the base model, this distribution may be normal or asymmetric, but the outliers tend to be of aleatoric character rather than epistemic. Figure 2 (top) shows a scatter plot for the predicted and actual values of the Software Effort test set with 255 regression models. We seem some outliers near 14000 for which models predict values between 4000 and 10000, leading to high residuals. This suggests that giving lower proportional weight to these errors in the loss function, as the $L_L$ loss in the bottom image does, may be a particularly interesting route to explore for assessors. This hypothesis is behind the experimental methodology in the following section.

## 3 LOSS FUNCTIONS AND PROBLEM REPRESENTATION

For the rest of the paper, base subjects $m_s$ are regression models $m_s : X \mapsto Y$, where $X \subset \mathbb{R}^d$ is an input feature vector and $Y \subset \mathbb{R}$ is the output. Given the output $\hat{y} = m_s(\mathbf{x})$ and the ground truth $y$, we can calculate any metric or loss function $L : \mathbb{R} \times \mathbb{R} \mapsto \mathbb{R}$, denoted as $L(\hat{y}, y)$. We will consider the following signed loss functions:

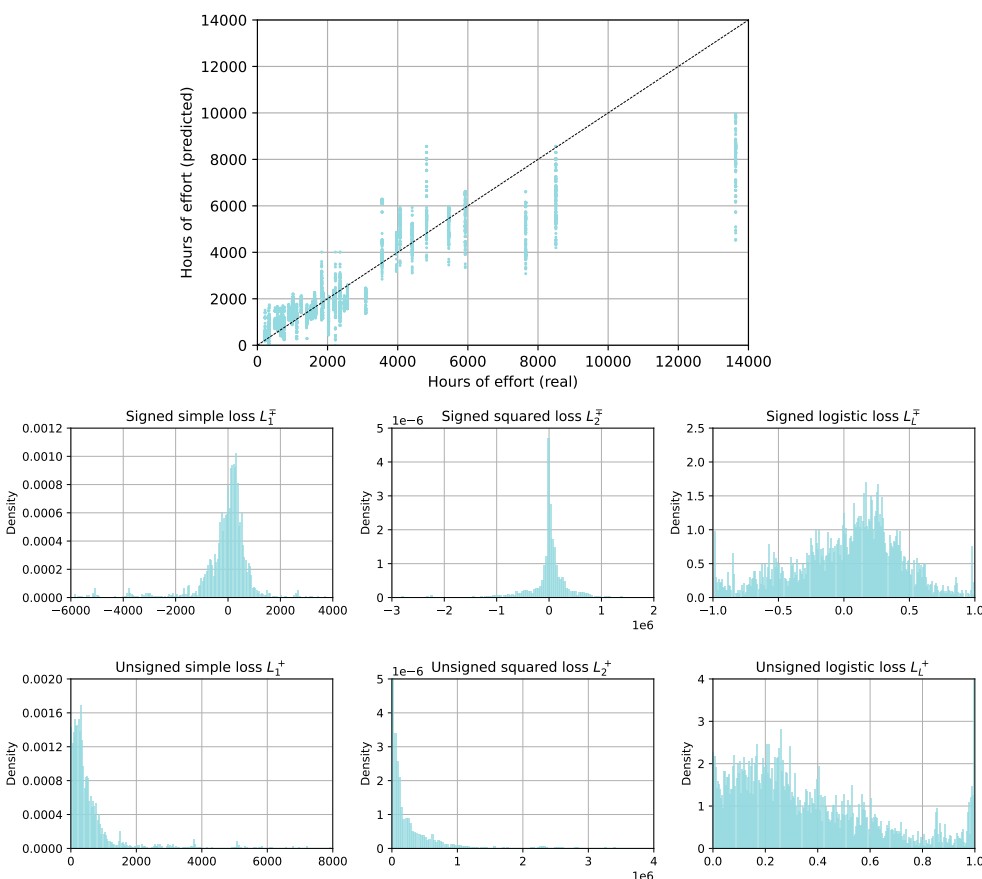

Figure 2: Software Effort dataset with 255 regression models. Top: scatter plot of $\hat{y}$ versus $y$. Bottom: histogram of losses, with first column corresponding to simple loss ($L_1^+$), second column to squared loss ($L_2^+$) and third column to logistic loss ($L_L^+$). The top and bottom rows represent the signed and unsigned versions, respectively. Assessors have to predict these losses. The shapes and the tails are very different.

**Definition 1** *Signed simple error*

$$L_1^{\mp}(\hat{y}, y) := \hat{y} - y \tag{1}$$

**Definition 2** *Signed squared error*

$$L_2^{\mp}(\hat{y}, y) := (\hat{y} - y) \cdot |\hat{y} - y| \tag{2}$$

**Definition 3** *Signed logistic error*

$$L_L^{\mp}(\hat{y}, y) := \frac{2}{1 + e^{-B(\hat{y}-y)}} - 1, B = \frac{\ln 3}{\text{mean}_Y |\hat{y} - y|} \tag{3}$$

The signed logistic error is a derivation from the general formula for a logistic curve so that values near $-1$ correspond to high underpredictions and values near $1$ correspond to high overpredictions. Additionally, since different regression tasks can have different ranges of errors (for instance, errors when predicting the number of rings in trees do not have the same magnitude as errors when predicting house pricings), we parametrise $L_L^{\mp}$ by a value $B$, such that the value of $L_L^{\mp}$ is 0.5 when the error in an instance is equal to the mean of the absolute errors of the base model.

The corresponding unsigned loss functions, are defined by simply removing the sign, i.e., $L_1^+ := |L_1^{\mp}|$, $L_2^+ := |L_2^{\mp}|$ and $L_L^+ := |L_L^{\mp}|$. It is easy to see that $L_1^{\mp}$, $L_2^{\mp}$ and $L_L^{\mp}$ are mononotically related

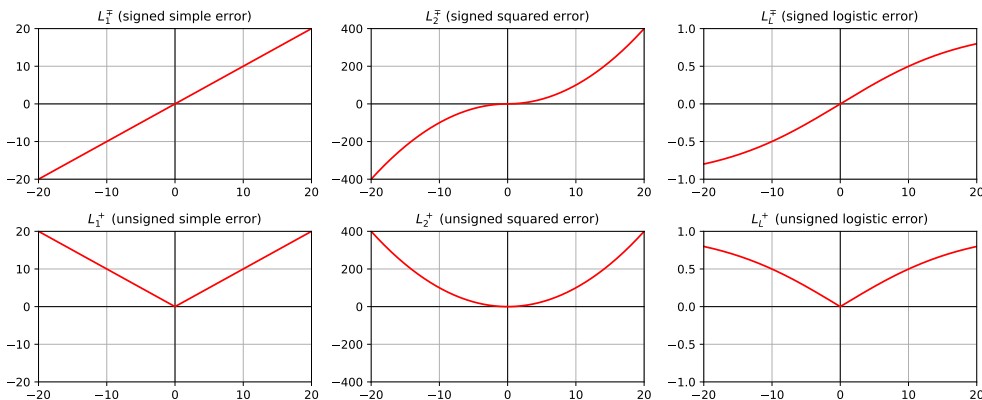

Figure 3: Functional representation of the six losses we use in this paper, signed ($L_1^{\mp}$, $L_2^{\mp}$ and $L_L^{\mp}$) and unsigned ($L_1^+$, $L_2^+$ and $L_L^+$).

(they do not lose information between each other), and the same happens between the unsigned versions. Of course, this no longer happens between the signed and unsigned versions, as the unsigned versions lose information. Figure 3 shows the six losses. The signed losses contain information about the *magnitude* and the *direction* of the error, whereas their unsigned counterparts only carry the *magnitude*, hence being less informative. The logistic loss tries to represent a smooth loss function that penalises outliers (mostly of aleatoric character) proportionally less than lower errors. It is hence a non-convex loss that, unlike the Huber Loss, does not fall in between the simple (linear) and squared errors, but goes beyond the linear error. It saturates on high residuals, but unlike Tukey's bi-weight loss, it is not piecewise, and has non-zero gradient everywhere (Tukey's loss is constant from a value, which is usually chosen to be 4.685 when residuals follow a standard normal distribution) (Belagiannis et al., 2015).

Once we have defined the loss functions, we must describe how to properly train assessors. Consider a class of subject systems $M$, which are represented by their size, number of parameters and other features, making a subject feature vector $\mathbf{m} \in M$. All these subject systems have previously been evaluated using a loss metric $L$. In order to build an assessor $a$, we need the input feature space $X$ and the subject space $M$ as inputs, and the loss as output, namely: $a : X \times M \mapsto \mathbb{R}$. The training set for the assessor is then composed of rows such as $\langle \mathbf{x}_i, \mathbf{m_s}, l_{i,s} \rangle$, where $i$ and $s$ are the instance and system indexes respectively, $l_{i,s} = L(\hat{y}_{i,s}, y_i)$ is the value to predict, with $y_i$ being the ground truth output for instance $i$, represented by $\mathbf{x}_i$ and $\hat{y}_{i,s} = m_s(\mathbf{x}_i)$.

In usual circumstances, $L$ is the target loss we care about and the one that appears in the training dataset for the assessor. However, in this paper we are going to distinguish between the target loss and the proxy loss. Consider that we build the training set $D_{tr}$ for the assessor with a proxy loss $L_{\circ\to}$, and we train the assessor $a$ for this loss. If the target loss, $L_{\to\circ}$, is different from the proxy loss, then we need to transform the output of the assessor $\hat{l}$ back to the target loss by using a transformation function $f$. This gives us two possible routes given a target loss $L_{\to\circ}$: we can either train an assessor that directly optimises for $L_{\to\circ}$ or train an assessor that optimises for a proxy error $L_{\circ\to}$ and then transform the assessor predictions via $f$. For instance, the transformation function $f$ between the unsigned simple error and the unsigned squared error is $f(l) = l^2$. Following the example of energy consumption from Figure 1, we could train an assessor model to predict the target loss (squared error) or train an assessor to predict a proxy loss (such as the unsigned logistic loss $L_L^+$) and then transform the output to obtain $L_2^+$.

## 4 METHODOLOGY AND EXPERIMENTAL SETUP

Training an assessor for a specific task requires *test results* from one or more base models. The more data and models we have the more the assessor can generalise. The quality of the assessor would also depend on the parametrisation of $\mathbf{x}$ and $\mathbf{s}$. In this regard, we have built a collection of base models as

a training resource for the assessor. We used 10 regression datasets of varying number of instances and attributes (see Table 1), as well as different distributions of the target variable. We use different *model configurations* (i.e., representing the combination of a model and its associated hyperparameters). Training such a model configuration on a specific dataset provides us instance-level results of the predicted and actual values on the test set, as well as additional metrics including training and inference time, and memory usage. These characteristics, paired with different hyperparameter values, define the model parametrisation $\mathbf{s}$.

Table 1: Summary of Datasets: number of features (#Feat.) and instances (#Inst.), the type of features they contain (categorical or numerical) and their domain.

| Dataset | #Feat. | #Inst. | Feat. Types | | Domain |
|---|---|---|---|---|---|
| | | | Cat. | Num. | |
| Abalone (Nash et al., 1995) | 8 | 4177 | ● | ● | Biology |
| Auction Verification (Ordoni et al., 2022) | 8 | 2043 | ● | ● | Commerce |
| BGN EchoMonts (Romano et al., 2021) | 10 | 17496 | ● | ● | Health |
| California Housing (Kelley Pace & Barry, 1997) | 8 | 20640 | ● | ● | Real State |
| Infrared Thermography Temp. (Wang et al., 2023) | 3 | 1020 | ● | ● | Health |
| Life Expectancy (World Health Organization, 2015) | 21 | 2938 | ● | ● | Health |
| Music Popularity (Kakkad, 2021) | 14 | 43597 | ● | ● | Music |
| Parkinsons Telemonitoring (*motor*) (Tsanas et al., 2009) | 20 | 5875 | | ● | Health |
| Parkinsons Telemonitoring (*total*) (Tsanas et al., 2009) | 20 | 5875 | | ● | Health |
| Software Cost Estimation (Hernández-Orallo, 2013) | 6 | 145 | ● | ● | Projects |

In order to have a homogeneous parametrisation $\mathbf{s}$ we train five distinct tree-based algorithms for each of the ten datasets. Specifically, we employed Decision Trees (Breiman et al., 1984), Random Forests (Ho, 1995), CatBoost (Prokhorenkova et al., 2019), XGBoost (Chen & Guestrin, 2016) and LightGBM (Ke et al., 2017). We explored up to 75 unique combinations of hyperparameter combinations: max depth values of 3, 5, 7, 9 and 11, learning rates of 0.01, 0.05 and 0.1, and 100, 250, 500, 750 and 1000 estimators. For decision trees, we used fewer configurations. Each dataset thus yielded a total of 255 different unique model variations (denoted by the system space $S$). We partitioned the data using a 70/30 train-test partition, and recorded the performance metrics at the instance level on the test set. Therefore, each row $\langle \mathbf{x}, \mathbf{s}, \hat{y}, y \rangle$ of the test set consists of a task instance representation $\mathbf{x}$ and a model configuration $\mathbf{s}$, with the corresponding predicted and actual results. These results serve as the training dataset for the assessors (link provided for final version).

The training process for the assessors is defined as follows: given a pair of target and proxy losses ($L_{\rightarrow \circ}$ and $L_{\circ \rightarrow}$, respectively), we train two assessors independently:

1. The *target assessor*: this assessor is trained to directly predict the target loss, using the tuple $\langle \mathbf{x}, \mathbf{s}, L_{\rightarrow \circ}(\hat{y}, y) \rangle$. No output post-processing is required.

2. The *proxy assessor*: this assessor is trained to predict the proxy loss $L_{\circ \rightarrow}$, using the tuple $\langle \mathbf{x}, \mathbf{s}, L_{\circ \rightarrow}(\hat{y}, y) \rangle$. The output is then transformed into the target loss, via the corresponding transformation function $f$.

The data for training the assessors is also partitioned using a 70/30 split, from the instance-level evaluation data set. This partitioning strategy is distinct from the initial split used for training the base models. Specifically, an assessor should not encounter, when predicting the test set, an example from the original problem $\mathbf{x} \in X$ that has been used to train said assessor, as this could produce contamination. This relies on keeping track of the instance identifier $x_{\text{id}}$. Several regression models were used as assessors: namely, XGBoost (Chen & Guestrin, 2016), linear regression (Galton, 1886), feed-forward neural networks (McCulloch & Pitts, 1943) and Bayesian ridge regression (Tipping, 2001), to account for the different strategies these models use to solve tasks (Fabra-Boluda et al., 2020; 2024), testing whether our results hold independently of the choice of assessor model.

In our analysis, we evaluate the relationship between the target and proxy assessors by calculating the Spearman's correlation coefficient $\rho$. To assess the statistical significance of the differences in $\rho$,s we establish 95% confidence intervals using a bootstrapping approach (Efron, 1979). We consider the differences between the proxy and target assessors statistically significant when these confidence

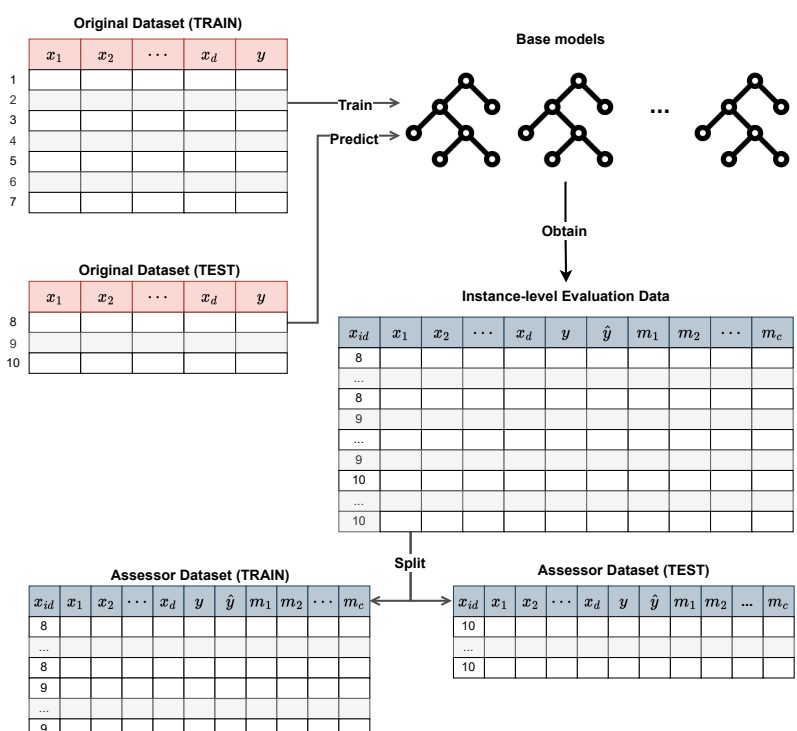

Figure 4: (Top) Procedure to obtain instance level evaluation results. In the final datasets, the original problem features $X$, as well as the model characteristics $S$, constitute an example for the assessor. (Bottom) To avoid contamination, a splitting method is applied to the data, so that the assessor training does not have any $\mathbf{x}$ that appears in the test for the assessor, with the same or different $\mathbf{m}$. The instance $\mathbf{x}$ identifier $x_{\text{id}}$ is only shown for illustration, but not used in the training or evaluation of the assessor.

intervals do not overlap. Furthermore, we quantify the performance of the proxy assessor relative to the target assessor by counting the number of datasets (out of the 10 in total) in which the proxy assessor achieves higher $\rho$ values. When the differences are not statistically significant, as indicated by overlapping confidence intervals, we categorise these cases as ties. This counting is formulated as the following score: $\#wins + \#ties + \#losses$, so that every win grants 1 point, every tie 0 points and every loss $-1$ points. Our score range goes from $-10$ (if the proxy assessor loses all 10 records) to 10 (if the proxy assessor wins all 10 records). A final aggregated score between $-1$ and 1 can be computed by obtaining the mean of these scores to assess the different approaches accounting for all datasets and all assessor model choices.

## 5 RESULTS

Figure 5 (left) shows the scores for all datasets when the assessor model of choice is XGBoost. Some interesting patterns can be seen: mainly, that learning from unsigned losses ($L_1^{\mp}$, $L_2^{\mp}$ and $L_L^{\mp}$) to predict their unsigned counterparts yields worse assessors than learning from $L_1^+$, $L_2^+$ and $L_L^+$ directly: for instance, when the proxy error is $L_2^{\mp}$ and the target error is $L_2^+$, the final score is $-9$ (e.g., from the 10 datasets, there is one tie – no significant differences in Spearman correlation – and 9 losses). This contrast is specially sharp with the simple signed error, where, in all 10 datasets, its absolute counterpart yields better results in terms of Spearman correlation $\rho$. Overall, the most underperforming proxy error is by far the signed squared error, managing scores between $-10$ and $-9$ (that means no wins at all), underperforming even when comparing it to other signed losses, indicating that it is not a good proxy loss to use in general.

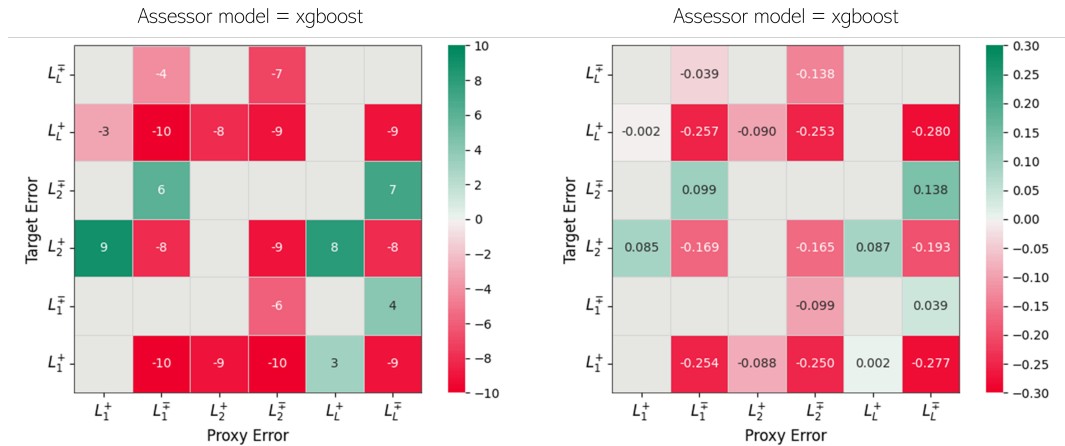

Figure 5: (Left) Score matrix for XGBoost assessor model. (Right) Aggregated Spearman margin matrix for XGBoost assessor model. In both matrices, rows represent target errors and columns proxy errors. Red values indicate poor performance from trying to predict $L_{\rightarrow\circ}$ by learning $L_{\circ\rightarrow}$. Inversely, green values show instances where learning from $L_{\circ\rightarrow}$ is better than from learning directly from $L_{\rightarrow\circ}$

One possible explanation for this under-performance is depicted in Figure 6: assessors with signed proxies (right plot) tend to make predictions closer to 0 (the mean), and the predictions (after the transformation $f$) underestimate the loss, even more so than those with unsigned proxies (left plot). This underestimation occurs for all the base models. For more details, see in Figure 13 in Appendix B.

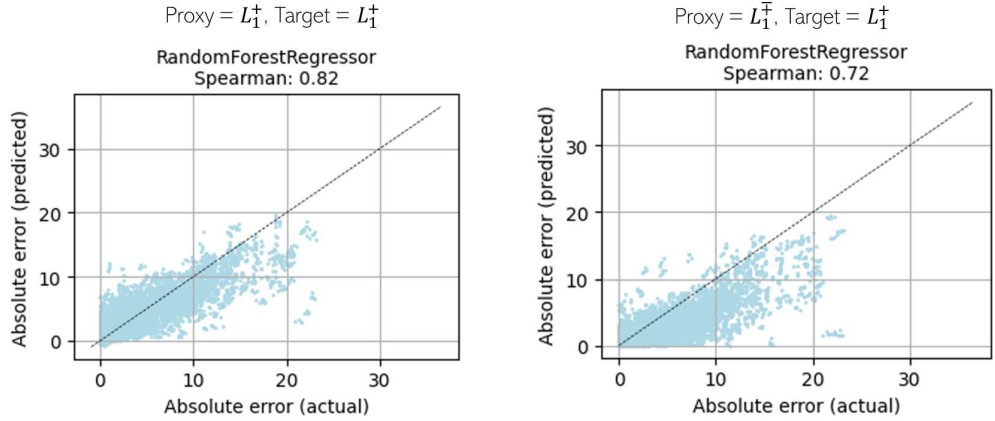

Figure 6: Scatter plots for the assessor of the Parkinson's Disease Rating Scale for RandomForestRegressor base models and assessor model XGBoost. Because the predictions of the assessor tend to the mean, the case where the proxy is signed takes predictions towards 0, and the predictions usually fall under the diagonal

In contrast, the logistic loss shows promising results: regarding $L_L^{\mp}$, when used as a proxy error to predict $L_1^{\mp}$ or $L_2^{\mp}$, it outperforms the target errors (4 and 7 points, respectively). A similar pattern can be seen with $L_L^{+}$, which obtains 3 and 8 points when used as a proxy error to predict $L_1^{+}$ and $L_2^{+}$, respectively. The simple unsigned error shows varying behaviour, outperforming $L_2^{+}$ but not being a good proxy to predict $L_L^{+}$.

These scores evaluate the performance of the approaches by counting the records where using a proxy loss is better than using the target loss directly. However, they are not able to quantify the

magnitude of said improvement. Figure 5 (right) addresses this, showing the mean Spearman difference of the 10 datasets for each combination of proxy and target error. For the computation of this mean, instances where $\rho$ is not significant are treated as having a difference equal to 0.

We see a similar behaviour to that depicted in the score matrix, although with some appreciations, specially regarding the logistic errors, where the differences are not as big as the scores matrix may suggest. The signed logistic loss presents the highest differences of the signed errors, although it manages to be a better proxy than the unsigned squared error.

These patterns are independent of the model chosen as assessor, as seen in Figure 7, where a mean score taking into account all datasets and assessor models is computed, resulting in values between $-1$ and $1$, with similar interpretation as when only analysing one assessor model. Equally, Spearman differences are computed for all datasets and assessor models, with similar patterns emerging in both matrices as the ones in Figure 5. See Appendix A to see score matrices of other assessor models.

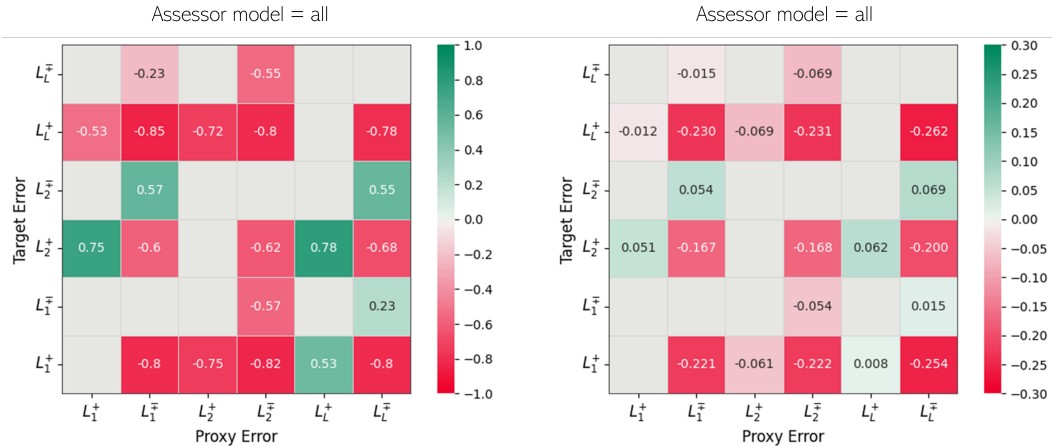

Figure 7: (Left) Mean score matrix of every possible approach between target and proxy errors. (Right) Aggregated Spearman margin matrix. In both matrices, rows represent target errors and columns proxy errors. Red values indicate poor performance from trying to predict $L_{\rightarrow \circ}$ by learning $L_{\circ \rightarrow}$. Inversely, green values show instances where learning from $L_{\circ \rightarrow}$ is better than from learning directly from $L_{\rightarrow \circ}$

Figure 8 summarises the results of this paper by comparing most of the pairs between target and proxy losses (shown in Spearman correlation margin). We can now see more clearly that the logistic loss wins over all the other losses in its column. There also appears to be some sense of transitivity between errors: for instance, training an assessor with the signed squared error as the proxy loss to predict the target loss unsigned simple error, there is a path (two paths, in fact), that say this proxy assessor would be worse than training directly with the target loss. As shown in Figure 7, this is correct. This property holds for all pairs of losses in the diagram. In cases where the arrows conforming a path are of different colours, the 'strength' of the arrows (differences in $\rho$, as shown in Figure 7) would dictate the final performance of the assessor.

## 6 DISCUSSION

AI assessors represent a second-order estimation problem whose goal is to predict a loss or utility function, for any new example and base subject model. This is much more flexible than uncertainty self-estimation because we can choose the metric of the assessor to be different from the ones the base models are optimised for or evaluated. Still, in this context it may seem natural to build an assessor to optimise for the target loss. However, we see that some other proxy losses may be more effective. Looking at the distribution of residuals, one explanation may be found in a double penalisation of high residuals (e.g., for outliers). That indicates that for convex loss functions used at the first-order level (base models) we may benefit for concave loss functions at the second level that compensate for the weight in the extremes of the distribution.

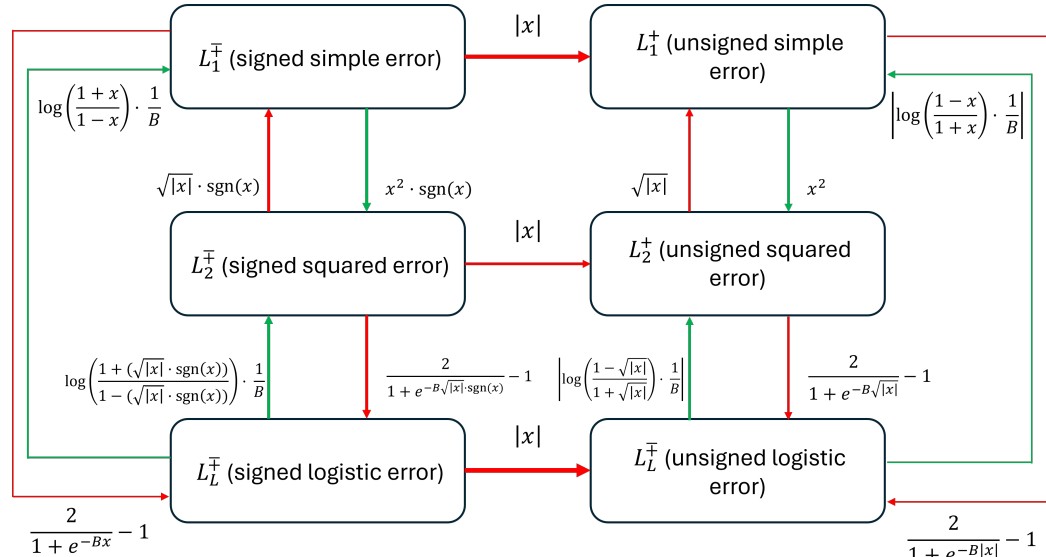

Figure 8: Which assessor metric to optimise? Signed and absolute versions of the same metric are arranged horizontally (the mapping is nonmonotonic, so only one direction is possible), while different metrics with monotonic transformations are arranged vertically. Arrows go from proxy metrics to target metrics. Green (respectively red) means the proxy metric is better (respectively worse) than the target metric when the target metric is to be optimised. The width of the arrow represents Spearman correlation margin. "Diagonal" transformations (for example, from signed simple error to unsigned squared error) are omitted for clarity, but shown in the matrices in figure 7

In this paper, we chose regression problems for this first analysis of proxy losses for assessors because loss functions for regression are well known, generally continuous, and the most common one, the squared error, augments the weight of the extremes. This suggests similar exploration for classification, and especially for losses in structured or generative tasks, Nocould be done following the methodology in this paper. Similarly, in situations where a metric is composed of several parts, e.g., components in a toxicity metric or precision and recall in the F1 score, it may make more sense to estimate the components (or some monotonic transformations of the components) with separate assessors and then integrate the prediction of the overall metric. Overall, this paper opens a wide range of options for exploring the impact of loss and utility metrics when building assessors.

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

## A    SCORE RESULTS FOR ALL ASSESSOR TYPES

This appendix contains more score and Spearman margin matrices, showing that the results obtained for XGBoost (the assessor model discussed in the main text) hold for more types of assessors:

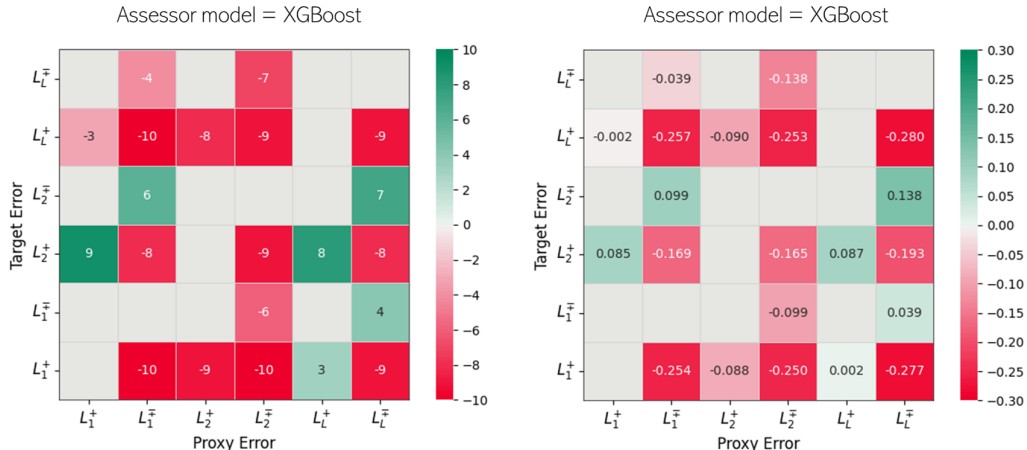

Figure 9: (Left) Score matrix for XGBoost assessor model. (Right) Aggregated Spearman margin matrix for XGBoost assessor model. In both matrices, rows represent target errors and columns proxy errors. Red values indicate poor performance from trying to predict $L_{\to\circ}$ by learning $L_{\circ\to}$. Inversely, green values show instances where learning from $L_{\circ\to}$ is better than from learning directly from $L_{\to\circ}$

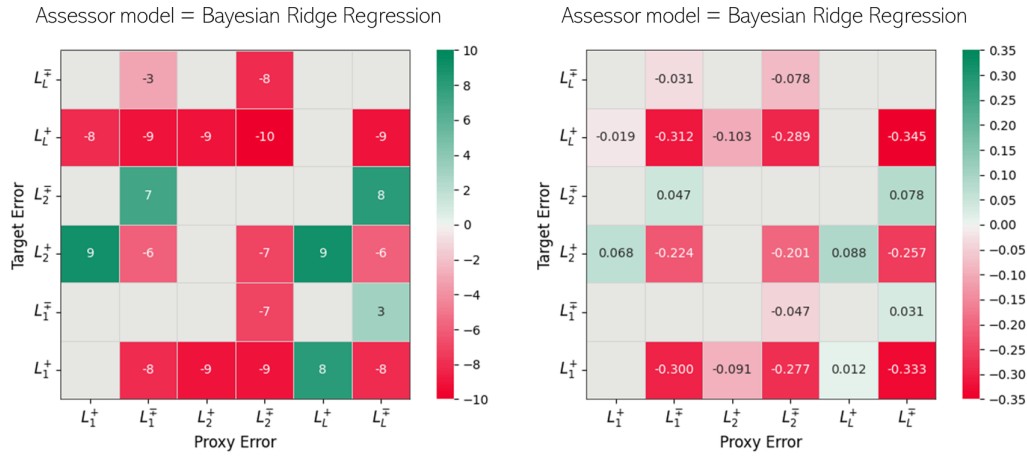

Figure 10: (Left) Score matrix for Bayesian ridge regression assessor model. (Right) Aggregated Spearman margin matrix for Bayesian ridge assessor model. In both matrices, rows represent target errors and columns proxy errors. Red values indicate poor performance from trying to predict $L_{\to\circ}$ by learning $L_{\circ\to}$. Inversely, green values show instances where learning from $L_{\circ\to}$ is better than from learning directly from $L_{\to\circ}$

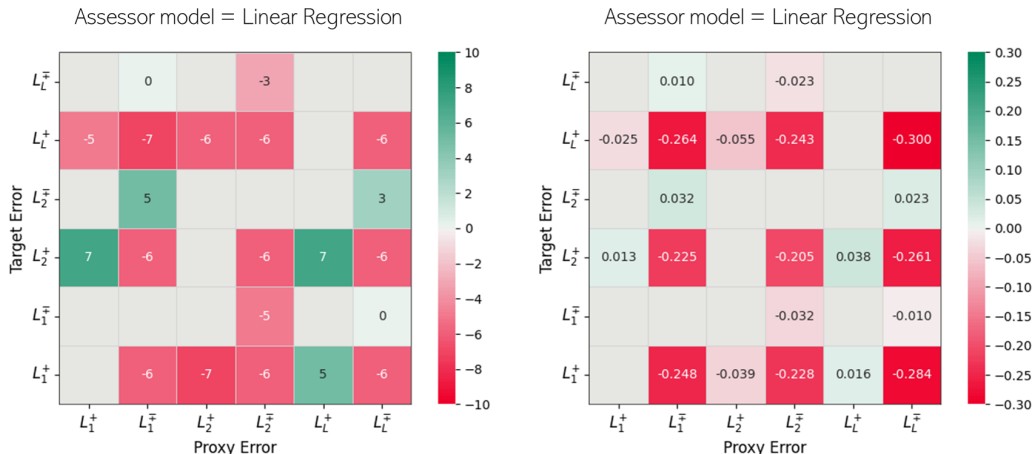

Figure 11: (Left) Score matrix for Linear Regression assessor model. (Right) Aggregated Spearman margin matrix for Linear Regression assessor model. In both matrices, rows represent target errors and columns proxy errors. Red values indicate poor performance from trying to predict $L_{\to\circ}$ by learning $L_{\circ\to}$. Inversely, green values show instances where learning from $L_{\circ\to}$ is better than from learning directly from $L_{\to\circ}$

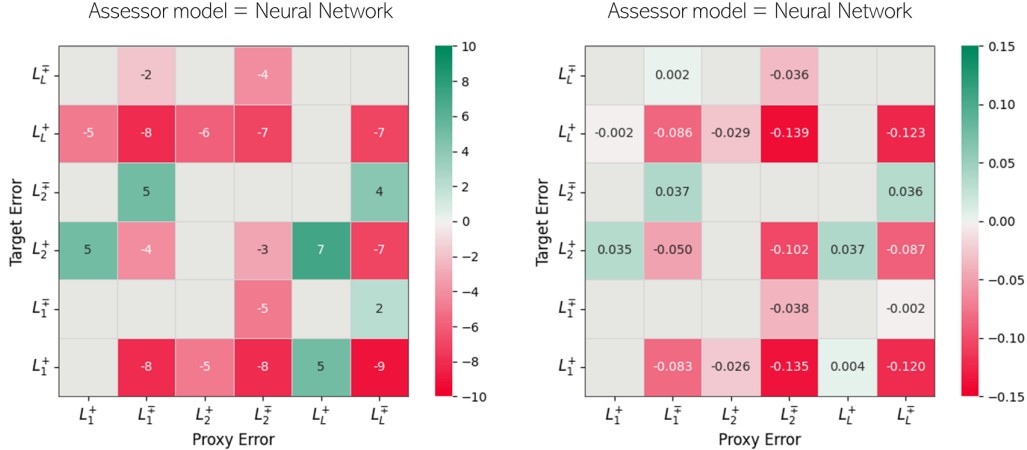

Figure 12: (Left) Score matrix for Feed-forward Neural Network assessor model. (Right) Aggregated Spearman margin matrix for Feed-forward Neural Network assessor model. In both matrices, rows represent target errors and columns proxy errors. Red values indicate poor performance from trying to predict $L_{\to\circ}$ by learning $L_{\circ\to}$. Inversely, green values show instances where learning from $L_{\circ\to}$ is better than from learning directly from $L_{\to\circ}$

Although the scores vary slightly (there are two groups with similar scores - XGBoost and Bayesian ridge regression vs Linear Regression and Neural Networks), the patterns are consistent: signed errors are not good proxies to predict their unsigned counterpants, and the logistic errors prove to be successful proxies.

## B  UNDERESTIMATION OF SIGNED ERRORS

Following the discussion on the main text (specifically, Figure 6), this section shows the full scatter plot for the XGBoost assessor on the Parkinson's Disease Rating Scale with different types of base models, as well as overall, when $L_1^{\mp}$ is used as proxy to predict $L_1^+$.

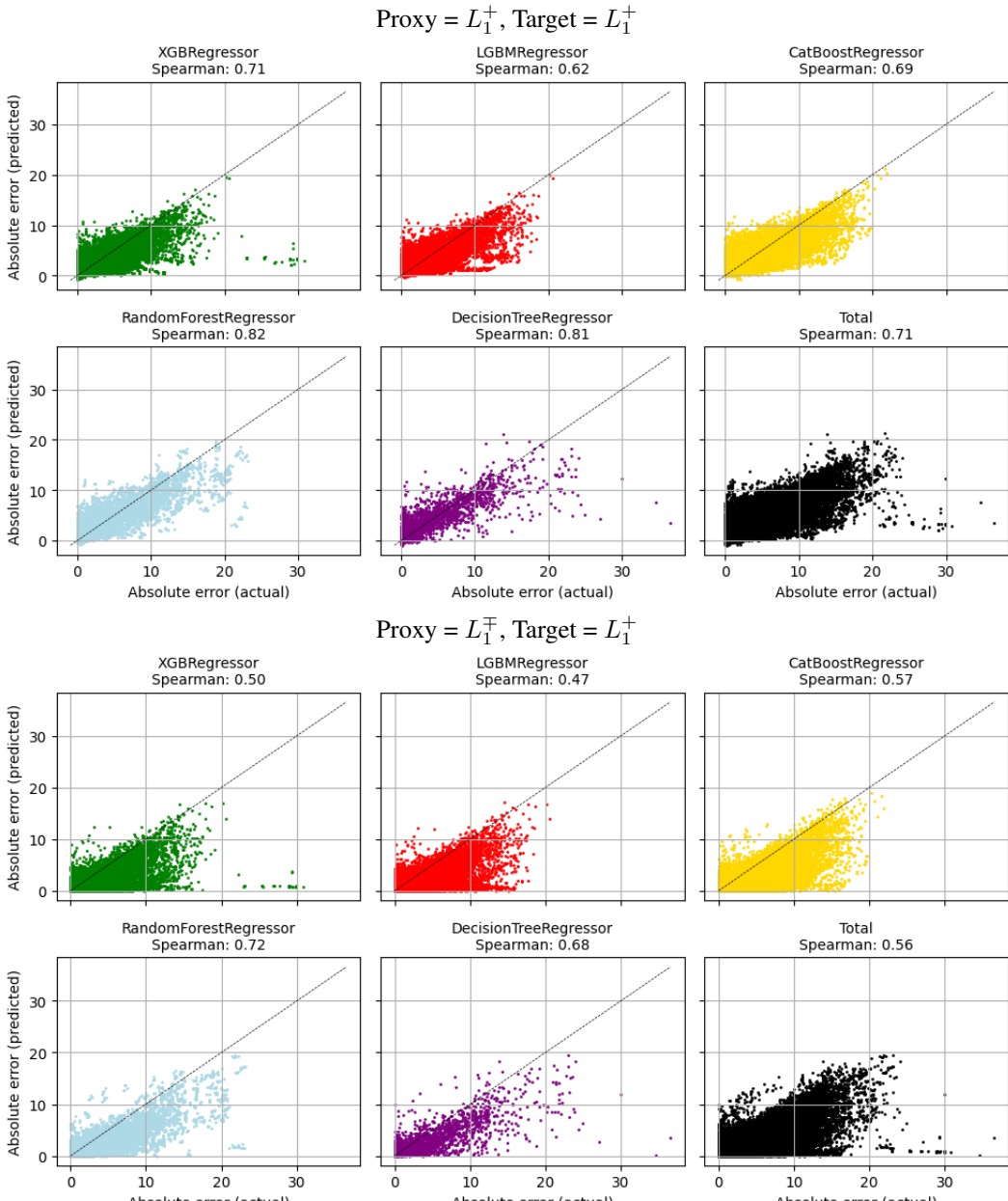

Figure 13: Scatter plots for the XGBoost assessor for the Parkinson's Disease Rating Scale and five base models: XGBRegressor, LGBMRegressor, CatBoostRegressor, RandomForestRegressor and DecisionTreeRegressor. Because the predictions of the assessor tend to the mean, the case where the proxy is signed takes predictions towards 0, and the predictions usually fall under the diagonal. This behaviour appears in all base models

