# OpenReview forum: "What should an AI assessor optimise for?"
_ICLR.cc/2025/Conference — Submitted to ICLR 2025_

### Official Review · Reviewer_rVaN · 2024-10-31

**Soundness:** 2
**Presentation:** 2
**Contribution:** 2
**Rating:** 3
**Confidence:** 4

**Summary:**

The paper investigates the question of what an AI assessor should optimize for. Specifically, could it be better to use something other than the target loss for the assessor. The question is considered empirically in the domain of tabular data using regression methods with a variety of methods of parameter optimization.

**Strengths:**

- The paper is generally clearly written.
- A fair number of experiments are conducted.

**Weaknesses:**

- The big weakness of the paper is that it is framed around assessing AI systems, but focuses on regression with tabular data. I don't see how the latter is relevant to the former.

**Questions:**

- Given the generality of the question, why are we focusing on tabular data?
- Line 39: please introduce the variable v.
- Line 47 "routing(Hu" missing space.
- I don't understand what Figure 1 is intended to illustrate. On the picture there are "target assessor" and "proxy assessor" but the caption text doesn't clearly define these. It would be helpful to see an example that illustrates the notion of better that the authors appear to be aiming for? Also, when might this be true?
- Why are the six losses chosen the right ones?
- If we are interested in AI broadly, why are we using decision trees, Random forests, etc for parameter optimization?
- It seems likely that they heuristic methods used for hyper parameter optimization or tabular data or regression might really be more of the story here than anything about AI.

---

> ### Author Response · Authors · 2024-11-27
> **Authors' rebuttal**
>
> We appreciate the comments from the reviewer. We now address the key points and questions:
>
> _**W1. Only regression and only tabular data**_
>
> We address the focus on regression and tabular data in Q1.
>
> _**Q1. Given the generality of the question, why are we focusing on tabular data?**_
>
> We chose tabular data for its simplicity in illustrating our framework, which can indeed be applied to any type of data, and also because it allows for a higher diversity of methods to treat the data and also for methods to build the assessors. For images, audio or text, for instance, almost every minimally effective technique is based on deep learning or is converted into a tabular representation through embeddings.
>
> We chose regression because it can suffer from the problem of outliers in the base prediction, a phenomenon that does not occur in problems that are evaluated with a binary or discrete metric (correct/incorrect, safe/unsafe). In many other areas, such as machine translation, summarisation, readability, etc., metrics are quantitative, and regression problems were then more appropriate to explore these quantitative scenarios.
>
> _**Q2. I don't understand what Figure 1 is intended to illustrate. On the picture there are "target assessor" and "proxy assessor" but the caption text doesn't clearly define these. It would be helpful to see an example that illustrates the notion of better that the authors appear to be aiming for? Also, when might this be true?**_
>
> Fig. 1 shows a possible use case for this work. Assume we have an energy consumption prediction model $M_{1}$ that we want to assess (i.e., predict in advanced the squared error of the model predictions, so we can act in accordance ㅡfor example, by ignoring $M_{1}$ and using a more powerful, albeit more expensive, model $M_{2}$ or to alert the user that the following prediction has a very large error and should be taken with caution). The usual way of doing this is by building an external predictor, called assessor, that tries to predict the squared error (in the figure, that would be the “target assessor”). We propose using another assessor (that would be the “proxy assessor” in the figure) that, instead of predicting the squared error directly, predicts another metric (let’s say, the logistic error) and then applying a transformation.
>
> Results show that, for instance, using a proxy assessor where the proxy metric is the unsigned logistic loss $L_{L}^{+}$ is better to predict the squared error $L_{2}^{+}$ base models make when predicting the Unified Parkinson's Disease Rating Scale (UPDRS), among others, than using a target assessor that predicts the squared error directly.
>
> _**Q3. Why are the six losses chosen the right ones?**_
>
> We chose these three loss functions (and their unsigned counterparts) because there is a monotonic transformation between all of them, and because they are frequently used in regression tasks. Other common losses, such as some logarithmic error in the predicted/actual quotient, were excluded because they are not based on the difference between the predicted and actual values. Still, other potential losses that deviate significantly from these monotonic relationships may be candidates for future research.
>
> _**Q4. If we are interested in AI broadly, why are we using decision trees, Random forests, etc for parameter optimization?**_
>
> We are interested in AI broadly, and ML more broadly than just deep learning. We used tabular data for simplicity. Then, the choice of tree-based models stems from their proven effectiveness in dealing effectively with tabular data. While we did explore neural networks, we found that XGBoost and similar methods often outperformed them with tabular input (this is consistent with https://doi.org/10.1016/j.inffus.2021.11.011). We wanted to show that our results generalise beyond simple scaling variables such as number of parameters or FLOPS. In fact, one of the main contributions of our work is that we extract scaling variables such as computing time that work across very different families of machine learning methods, and are predictive about performance. This brings generalisability to situations where other factors, such as deployment compute, may be relevant, and an independent (and non-linear) assessor model can replace a simple scaling law (https://aclanthology.org/2024.scalellm-1.1/).
>
> _(Given the character limit, we address the last question in the following comment)_

---

> > ### Author Response · Authors · 2024-11-27
> > **Authors' rebuttal: Q5**
> >
> > _**Q5. It seems likely that they heuristic methods used for hyper parameter optimization or tabular data or regression might really be more of the story here than anything about AI**_
> >
> > Externally anticipating the performance of a machine learning model is a core problem in AI. Just because the models are tabular does not make this less machine learning or less AI, especially when the methodology can be easily extrapolated to any other task in AI (including non-machine learning ones, such as constraint satisfaction, SAT problems, etc.). We made our choices for the experimental setting to remove confounders, and in some ways the diversity of techniques that we used suggests there is more generalisation than if we had only used transformers, as many papers do.

---

### Official Review · Reviewer_nvKT · 2024-11-03

**Soundness:** 2
**Presentation:** 2
**Contribution:** 2
**Rating:** 5
**Confidence:** 3

**Summary:**

In this paper the authors investigate the question as to whether assessor models should be trained using a proxy metric rather than the metric of interest. In doing so, they find mixed results. They also demonstrate that training assessors using more informative losses is not always better than training with uninformative versions.

**Strengths:**

I think the paper has several strengths:
1. The explanations of concepts and methodology is quite clear. I like the use of diagrams such as Figures 1, 3, 4 and 8.
2. The number of datasets used is quite commendable. 10 seems like a sufficient number to generate some useful trends if there was a large effect at play.
3. The subject matter is of interest although I think the focus on tree-based methods restricts the utility of the paper.

**Weaknesses:**

I do not think this paper is sufficiently novel and broadly applicable in order to warrant publicaton in ICLR.

I believe there are several weaknesses:
1. Unless I missed something, I believe all models used in the training set for the assessors are tree-based and are not neural networks. I know that tree-based methods can be better for tabular data but the most interesting application of assessors (to me at least) is neural networks. The most interesting assessor models are the ones that produce scaling laws.
2. The results of the paper do not seem particularly surprising. We know that changing the geometry of the loss landscape by using a different loss function can sometimes improve pefromance in regression. I don't think applying this logic to assessor models is paticularly interesting especially given the limited scope of exploration of the paper to tree based methods and the lack of mathematical theory.
3. Outside of intuition, there is no theory developed to explain the results. It seems that for a paper at ICLR, I would have expected more here.

**Questions:**

There may be some assumptions in my review which are incorrect. Please correct me if I am wrong but I am assuming that:
1. There are no neural network models which the assessors are applied to but neural networks are used to do asessing.
2. There are no proofs or more formal justifications for the relationships observed

I have a few other questions / requests:
1. Could the authors replace the figures with PDFs or SVGs intead of what is currently being used (easier to zoom in on)?
2. Do the authors think that there might be different relationships in the case of assessors applied to predicing things about over-parameterised models?

---

> ### Author Response · Authors · 2024-11-27
> **Authors' rebuttal**
>
> We appreciate the review. We now address some of the points mentioned:
>
> _**W1. Only tree-base models, not neural networks**_
>
> The choice of tree-based models stems from their proven effectiveness in dealing effectively with tabular data. While we did explore neural networks, we found that XGBoost and similar methods often outperformed them with tabular input (this is consistent with https://doi.org/10.1016/j.inffus.2021.11.011). We wanted to show that our results generalise beyond simple scaling variables such as number of parameters or FLOPS, which are well-known in deep learning but not in other machine learning methods. Actually, one of the main contributions of our work is that we extract scaling variables such as computing time that work across very different families of machine learning methods, and are predictive about performance. This brings generalisability to situations where other factors, such as deployment compute, may be relevant, and an independent (and non-linear) assessor model can replace a simple scaling law (https://aclanthology.org/2024.scalellm-1.1/)
>
>
> _**W2. Given limited scope of tree models, results are not surprsising**_
>
> The results are surprising in that optimising for the target metric is not optimal for the assessor, and further justifies decoupling the uncertainty or error estimation from the base model using assessors. We don’t think this finding is associated with tree models.
>
>
> _**W3. Lack of theoretical explanation**_
>
> We acknowledge the need for more theoretical explanation and aim to include this in future extensions of our paper, considering the distribution of the errors.
>
>
> _**Q1. There are no neural network models which the assessors are applied to but neural networks are used to do assessing. Is this correct?**_
>
> Yes, this is correct, one of the assessor models was a feed-forward neural network. Assessor models are machine learning models, and they do not need to resemble the base models.
>
>
> _**Q2. There are no proofs or more formal justifications for the relationships observed. Is this correct?**_
>
> We acknowledge the need for more theoretical groundwork and aim to include this in future extensions of our paper, considering the error distribution.
>
>
> _**Q3. Could the authors replace the figures with PDFs or SVGs intead of what is currently being used (easier to zoom in on)?**_
>
> We will ensure that in future versions our figures are not bitmaps.
>
>
> _**Q4. Do the authors think that there might be different relationships in the case of assessors applied to predicting things about over-parameterised models?**_
>
> Assessors are inherently designed to learn from test results, regardless of the underlying characteristics of the base models, including whether they are over-parameterised. For instance, if a model is poorly generalised and performs poorly on test data due to issues such as memorisation of the training set (because of over-parametrisation), the assessor can learn from these test results and adjust its predictions accordingly, even determining the areas where it will not generalise well. Consequently, the role of the assessor remains consistent, focusing on the observed performance without being directly influenced by the parameterisation of the base model. Of course, it may be affected by limited test data and over-parametrisation of the assessor itself.

---

### Official Review · Reviewer_HkoA · 2024-11-04

**Soundness:** 3
**Presentation:** 2
**Contribution:** 2
**Rating:** 5
**Confidence:** 4

**Summary:**

The paper studies the effects of training different prediction models on proxy assessors instead of the target metric. The study focuses on regression problems and tabular features. Results show that training for more informative losses is not always better. In addition, the paper provides a decomposition of results on different directions of pairwise combinations of loss functions.

**Strengths:**

S1: The exploration of what loss functions are most effective for a target lost is generally important and can be informative if best choices are known and documented for practitioners.

S2: The work uses a variety of tabular models and datasets.

**Weaknesses:**

W1: The work engages only with regression models, which makes the scope somewhat narrow.

W2: The work does show some interesting outcomes, with some of them summarized in Figure 8. The results are however exclusively explained from an empirical perspective only. There is no theoretical discussion about for example how ML optimization and different optimization algorithms may interfere and interact with the results. For example, the fact that signed simple error -> signed squared error is green (proxy easier), may be explained by optimization difficulties. For this one in particular, it would be useful to experiment with different optimization algorithms and settings of such algorithms. It would also be useful to understand whether things like learning rate etc have an impact on results.

**Questions:**

Q1: Are the artifacts of this work going to be made available to the public (e.g. code)?

Q2: Are there any findings or insights that are specific to some of the datasets or do all results apply to all datasets individually? It would be useful to have some breakdown per datasets and see if there are any characteristics of the data that strengthen the results (or otherwise make them less applicable).

Q3: Which optimization algorithms did you use?

---

> ### Author Response · Authors · 2024-11-27
> **Authors' rebuttal**
>
> We thank the reviewer for their comments. We respond to some of the comments first and then answer the questions:
>
> _**W1. Narrow scope for only working with regression**_
>
> We focused on regression metrics because they are particularly sensitive to outliers in the base predictions, a challenge that is less pronounced in problems evaluated using binary or discrete metrics (e.g., correct/incorrect, safe/unsafe). Additionally, many domains, such as machine translation, rely on quantitative metrics to evaluate performance. This makes regression a more appropriate framework for investigating scenarios where outcomes are inherently numerical.
>
>
> _**W2. Lack of theoretical explanation**_
>
> We acknowledge the need for more theoretical groundwork and aim to include this in future extensions of our paper, as well as experimenting with different optimisation algorithms.
>
>
> _**Q1. Are the artifacts of this work going to be made available to the public (e.g. code)?**_
>
> Yes, code, data employed and instance level data produced (as in compliance with https://doi.org/10.1126/science.adf6369) will be made public when accepted.
>
>
> _**Q2. Are there any findings or insights that are specific to some of the datasets or do all results apply to all datasets individually? It would be useful to have some breakdown per datasets and see if there are any characteristics of the data that strengthen the results (or otherwise make them less applicable)**_
>
> Our analysis suggests that the results are generally consistent across datasets.  We will provide the dataset-level results in the Appendix.
>
>
> _**Q3. Which optimization algorithms did you use?**_
>
> All base models and assessors used default hyperparameters. Preliminary experiments using grid search to find the best hyperparameters for the assessors showed consistent results, so we decided not to optimise the assessors further. This decision was made to keep the experimental setup as simple and computational lightweight as possible (given the dozens of thousands of runs performed).

---

### Official Review · Reviewer_N3pR · 2024-11-04

**Soundness:** 3
**Presentation:** 2
**Contribution:** 3
**Rating:** 5
**Confidence:** 3

**Summary:**

This paper explores the efficacy of training assessor models, which are predictive systems designed to estimate the performance of other models (base systems) on various metrics before deployment. The authors question the common assumption that directly optimizing assessors for a specific validity metric always leads to the best outcomes for that metric. They investigate this through a series of experiments involving regression problems, comparing the performance of assessors trained on different error metrics, including absolute error, squared error, and logistic error. The results reveal that optimizing for a proxy metric can sometimes outperform direct optimization for the target metric, with logistic error frequently yielding the best performance.

**Strengths:**

1. **Innovative Approach:** The paper addresses a critical gap in the understanding of how assessors can be trained, providing a fresh perspective on the relationship between the optimization of assessor models and the performance metrics of interest.
2. **Comprehensive Experimental Design:** The authors conduct several experiments that systematically evaluate the impact of different training metrics on assessor performance, leading to insightful findings.
3. **Interesting Findings:** The consistent superiority of logistic error in various situations prompts further investigation into the characteristics of error distributions, which is a valuable contribution to the field.

**Weaknesses:**

1. **Limited Metric Scope:** While the paper examines several regression metrics, it may benefit from a broader exploration of other types of metrics across different applications, such as classification or ranking tasks, to enhance the generalizability of the findings.
2. **Theoretical Foundation:** The paper would be strengthened by a more robust theoretical explanation of why certain metrics outperform others in the context of assessor training, particularly for the surprising results related to logistic error.
3. **Clarification Needed on Assessor Application:** The practical applications of assessors in real-world scenarios could be discussed in greater detail, including how to implement these findings in various domains, such as healthcare or autonomous systems.

**Questions:**

- Can the authors elaborate on the specific characteristics of the logistic error that made it the most effective metric across the experiments?
- How might the findings change if applied to more complex models or different domains? Are there limitations in the current experimental setup that could affect generalizability?
- It would be beneficial for the authors to provide concrete examples of how their findings could inform the design of assessor models in practice.

---

> ### Author Response · Authors · 2024-11-27
> **Authors' rebuttal**
>
> We appreciate the reviewer’s comments. We first address the points raised and then answer the questions:
>
> _**W1. Limited Metric Scope**_
>
> We focused on regression metrics because they are particularly sensitive to outliers in the base predictions, a challenge that is less pronounced in problems evaluated using binary or discrete metrics (e.g., correct/incorrect, safe/unsafe). Additionally, many domains, such as machine translation, summarisation, readability, etc., rely on quantitative metrics to evaluate performance. This makes regression a more appropriate framework for investigating scenarios where outcomes are inherently numerical.
>
> _**W2. Theoretical Foundation**_
>
> We acknowledge the need for more theoretical groundwork, such as particular error distributions for which we can show the effect of double penalisation. We left  this for future extensions of our paper.
>
> _**W3. Clarification Needed on Assessor Application**_
>
> We address the practical applications of assessors in Q3.
>
>
> _**Q1. “Can the authors elaborate on the specific characteristics of the logistic error that made it the most effective metric across the experiments?”**_
>
> The main characteristic of the logistic error is that, contrary to the other error functions, it is concave. Since the assessor models are optimised to minimise the mean squared error of the assessor train dataset, we believe that there is some kind of “compensation”: when using convex metrics, such as the squared error, for the base regression model. High errors are “doubly penalised”, i.e., once from the base regression error metric and another for the optimisation metric of the assessor, which is always the squared error. This makes assessors less effective at learning the error distribution, especially when there are outliers, since the focus is put on those outliers. Concave functions like the logistic error mitigate this effect (hence the “compensation”), helping assessors to achieve better results overall.
>
>
> _**Q2. “How might the findings change if applied to more complex models or different domains? Are there limitations in the current experimental setup that could affect generalizability?”**_
>
> Our findings suggest that the current setup is adaptable to different complex models and different domains, as long as the base models  provide some form of prediction (be it a real value, a vector of probabilities, an action…) and a corresponding outcome metric (difference error, correct/incorrect label, Brier score, safety measure…) The distribution of the outcomes for this metric is what needs to be modelled by the assessor and appropriate transformations could be introduced in each of these cases. For instance, if what matters is a safety metric that heavily penalises some types of outcomes, we may get better results by doing a logarithmic transformation of that metric before training the assessor. A case-by-case analysis may be required but the lessons learned from this paper should be generally applicable to these other scenarios.
>
>
> _**Q3. “It would be beneficial for the authors to provide concrete examples of how their findings could inform the design of assessor models in practice”**_
>
> Our findings can inform the design of practical assessor models by using logistic error or other concave functions to improve training. These metrics help models perform well with different error distributions. For example, in healthcare, an assessor could predict recovery time estimation success by considering patient factors without overfitting for rare cases. In autonomous systems, models could predict safety outcomes by scoring errors without heavily penalising rare but important deviations.

---

### Author Response · Authors · 2024-11-27
**Authors' rebuttal**

We are thankful for the comments received but if these scores are kept we will likely withdraw the paper in order to fully implement the suggested changes (i.e., increase the scope to other kinds of tasks and methods, include dataset-level results and develop some theoretical analysis). We could do some of them for the camera ready deadline, but not during the rebuttal period.

---

### Meta-Review · Area_Chair_W6Uk · 2024-12-21

**Metareview:**

This paper investigates whether AI assessors should optimize directly for a target loss or a proxy loss with a transformation to the target loss. The authors conduct experiments on regression tasks with tabular data, comparing various loss functions, including squared error, absolute error, and logistic loss. The key finding is that optimizing for proxy losses, particularly logistic loss, can yield better performance than direct optimization, challenging conventional assumptions.

**Strengths**
- The paper poses an innovative and practically relevant question regarding the optimization of assessor models
- Comprehensive experimental evaluation across ten datasets
- Interesting findings showing that logistic loss can serve as an effective proxy for assessor models

**Weaknesses**
- The study is limited to regression tasks with tabular data (with restricted generalizability)
- The paper lacks theoretical explanations
- The choice of tree-based models for assessors limits applicability in more general scenarios
- The practical relevance of the findings in real-world scenarios needs further elaboration.

In summary, while the paper presents an interesting and well-motivated problem with promising empirical findings, its limited scope, lack of theoretical insights, and restricted applicability weaken its overall impact.

**Additional Comments On Reviewer Discussion:**

The reviewers acknowledged the paper's innovative question and comprehensive experiments but noted concerns about its limited scope, lack of theoretical grounding, and restricted applicability of tree-based models.

---

### Decision · Program_Chairs · 2025-01-22

Reject